

# HOAPS and ERA-Interim precipitation over sea: Validation against shipboard in-situ measurements

Karl Bumke[1], Gert König-Langlo[2], Julian Kinzel[3], and Marc Schröder[3]

[1] GEOMAR Helmholtz Centre for Ocean Research Kiel, Kiel, 24105, Germany
[2] Alfred Wegener Institute Helmholtz Centre for Polar and Marine Research, Bremerhaven, 27570, Germany
[3] Deutscher Wetterdienst, Satellite Based Climate Monitoring, Offenbach, 63067, Germany

*Correspondence to*: K. Bumke (kbumke@geomar.de)

**Abstract.** The satellite derived HOAPS (Hamburg Ocean Atmosphere Parameters and Fluxes from Satellite data) and ECMWF (European Centre for Medium-Range Weather Forecasts) ERA-Interim reanalysis data sets have been validated against in-situ precipitation measurements from ship rain gauges and optical disdrometers over the open-ocean by applying a statistical analysis for binary forecasts. For this purpose collocated pairs of data were merged within a certain temporal and spatial threshold into single events, according to the satellites' overpass, the observation and the forecast times. HOAPS detects the frequency of precipitation well, while ERA-Interim strongly overestimates it, especially in the tropics and sub-tropics. Although precipitation rates are difficult to compare because along-track point measurements are collocated with areal estimates and the numbers of available data are limited, we find that HOAPS underestimates precipitation rates, while ERA-Interim's Atlantic-wide average precipitation rate is close to measurements. However, regionally averaged over latitudinal belts, there are deviations between the observed mean precipitation rates and ERA-Interim. The most obvious ERA-Interim feature is an overestimation of precipitation in the area of the intertropical convergence zone and the southern sub-tropics over the Atlantic Ocean. For a limited number of snow measurements by optical disdrometers it can be concluded that both HOAPS and ERA-Interim are suitable to detect the occurrence of solid precipitation.

## 1 Introduction

Precipitation is one of the key parameters of the global water cycle. Therein the precipitation over the ocean is especially important since it contributes more than 75 % of the global annual total precipitation (Schmitt, 2008). Due to climate change it is reasonable to expect changes in both the pattern and amount of precipitation (Liu and Allan, 2013, O'Gorman et al., 2012 and Trenberth, 2011), as indicated by changes in the horizontal surface salinity distribution over the ocean (Durack et al., 2012). However, until now it has not been possible to detect robust trends in regional precipitation over the ocean because in-situ precipitation measurements are sparse and uncertain (Rhein et al., 2013). The progress in satellite technology has provided the possibility to retrieve global data sets from space, including precipitation. In addition to the Hamburg Ocean Atmosphere Parameters and Fluxes from Satellite data (HOAPS, Andersson et al., 2010 and 2011)), remote sensing based precipitation data sets are alternatively available e.g. from the Global Precipitation Climatology Project (GPCP, e.g.





Huffman et al., 1997), but concerns have been expressed over the need for further work to evaluate these data sets (Rhein et al., 2013). Recent progress in reanalyses shows improved accuracy in precipitation (Dee et al., 2011), although tests for internal consistency among different components of the hydrological cycle in reanalysis data still reveal some issues (Trenberth et al., 2011). This is supported by a study of Andersson et al. (2011), who pointed out that state-of-the-art satellite

retrievals and reanalysis data sets still disagree on global precipitation amounts, patterns, variability, and temporal behaviour, with the relative differences increasing in the poleward direction. Validation studies so far have been done mostly against measurements on atolls (e.g. Pfeifroth et al., 2013), although the extent to which atoll gauges are representative of open-ocean rainfall is still an open question (e.g. Wang et al., 2014). In the present study we use in-situ ship rain gauge (Hasse et al., 1998) and optical disdrometer (ODM 470) data (Großklaus et al., 1998), gained on board of research vessels, to validate

two data sets, the HOAPS (Andersson et al., 2010 and 2011) and the ECMWF (European Centre for Medium-Range Weather Forecasts) ERA-Interim reanalysis (Dee et al., 2011) data set.

Section 2 gives an overview over the data and instruments used. The collocation of the data is described in section 3, followed by an overview of used validation methods in section 4. In section 5 we present our results, followed by a discussion of the results and the summary and outlook in sections 6 and 7.

**2 Data**

For this study data from two measurement periods, 1995 to 1997 and 2005 to 2008 are available.

The 1995 to 1997 data were collected on the German ship R/V Meteor using a ship rain gauge (SRG, Hasse et al., 1998), and on board the German R/V Polarstern, the US R/V Knorr, and the US R/V Ron Brown using optical disdrometers (ODM 470, Großklaus et al., 1998). During this period the ODM 470s were coupled with opto-electronic infrared rain sensors, switching

the sensors on with the onset of precipitation, and switching them off about 30 minutes after the last precipitation occurrence. Due to reported sporadic male-functions of the rain sensor we use only periods, where data of the ODMs are available and did not assume that there was generally no precipitation along the track when ODMs were switched off. Measurements are available at 8 minute intervals. Data were collected over the Atlantic Ocean and the tropical Pacific Ocean, positions of collocated data are shown in Fig. 6.

For 2005 to 2008, SRG data are available, collected on board of the German R/V Polarstern  and R/V Maria S. Merian (Bundesamt für Seeschifffahrt und Hydrographie, 2015). A description of the meteorological observatory of Polarstern can be found in König-Langlo et al. (2006). In contrast to the earlier period, the instruments were run continuously. Here the used measurement intervals are 10 minutes. Data are collected mostly over the Atlantic Ocean; positions of collocated data can be seen from Fig. 6.



## 2.1 Ship rain gauge measurements

The SRG is commercially available from Eigenbrodt Environmental Measurement Systems near Hamburg, Germany. An outstanding feature of the SRG is an additional lateral collector, which is effective especially under high wind speed conditions (Hasse et al., 1998). Collected water, estimated separately from the top and lateral collector, in combination with

measured wind speeds relative to the instrument allows the derivation of true rainfall rates (Clemens, 2002). Comparisons to other instruments show that the SRG performs well and gives nearly unbiased estimates of rainfall (Clemens and Bumke, 2002).

Position and time at the end of each measurement interval were taken from the ship's measurement system. Measurement intervals were set to 8 min for R/V Meteor (1995-1997), while R/V Polarstern data (2005-2008) are available for 10 min

intervals. R/V Maria S. Merian data (2007-2008) have been integrated over 10 min, based on 1 min measurement intervals. Since SRGs are not suitable to measure snow, only data collected at air temperatures above 4° C have been used. Long term measurements on the main building of the GEOMAR Helmholtz Centre for Ocean Research Kiel, Germany, have shown that nearly all discrepancies between ODM 470 measurements, which give extreme high precipitation rates using the rain algorithm in case of solid precipitation, and SRG measurements, occur at temperatures below 4°C, which is in agreement

with a study of Froidurot et al. (2014).

## 2.2 Shipboard ODM 470 optical disdrometer measurements

The ODM 470 (Großklaus et al., 1998) is also commercially available from Eigenbrodt Environmental Measurement Systems. It was successfully validated to measure precipitation even under strong wind conditions (Großklaus, 1996; Bumke et al., 2004). Lempio et al. (2007) further developed the snowfall algorithm and applied it to ODM 470 measurements during

an inter-comparison field campaign in Uppsala, Sweden during winter 1999/2000. Comparison with gauge data and daily manual measurements showed reliable instrument performance. The correlation coefficient to 56 days of manual measurements was r=0.794.

The measurement principle of the ODM 470 is light extinction caused by hydrometeors passing through a cylindrical sensitive volume, which is kept perpendicular to the local wind direction with the aid of a wind vane. The local wind speed is

measured using a cup anemometer. The disdrometer measures the size of the cross-sectional area and the residence time of hydrometeors in the sensitive volume, within a used size range in diameter of 0.4–22 mm (snow version) and 0.5-6.4 mm (rain version). Measurements are partitioned into 128 size bins; with the highest resolution at small particles and a logarithmic increase in size (snow version) or constant resolution and a linear increase in size (rain version). Coincidence effects of multiple hydrometeors within the sensitive volume at the same time, and edge effects of partly scanned

hydrometeors, are considered in the same way for liquid and solid phase precipitation. The precipitation rate in mm h$^{-1}$ is calculated using the size bins, terminal velocity, mass of the hydrometeors, and local wind speed (Clemens, 2002). The determination of the rainfall rate through the liquid water content (mass) and the fall velocity is easily parameterized, as rain



drops have a nearly spherical shape and constant density. In contrast to rainfall, solid precipitation is characterized by a variety of complex shapes with different fall velocities and different equivalent liquid water contents. The measured cross-sectional area depends on the size, shape and orientation of the solid particles hindering the development of a unique solid precipitation retrieval scheme. The relationship between mass or equivalent liquid water content and the terminal fall

velocity for snow crystals was analysed by Hogan (1994) as a function of their maximum dimension. However, the disdrometer measures the size of the cross-sectional area instead of the maximum dimension of non-spherical particles. Assuming that the ice crystals fall randomly oriented through the sensitive volume (Brandes et al., 2007), Lempio et al. (2007) found from theoretical experiments using a ray tracing model (Macke et al., 1998) that the products of the terminal velocity and the equivalent liquid water content, as a function of the cross-sectional area, for different types of snow crystals

are of the same order of magnitude. That allows to use one common parameterization for all kinds of crystals, for practical reasons lump graupel was chosen. As it is nearly spherical in shape, it needs no transformation function from cross-sectional area to maximum dimension. The parameterization for lump graupel, which was the most frequently observed precipitation type over the Nordic Seas during the LOFZY campaign (Klepp et al., 2010), is applicable for particles with a size range of 0.4–9 mm.

The snow version of the ODM 470 was mounted on R/V Knorr and R/V Polarstern, the rain version on R/V Ron Brown. While all measurements on R/V Knorr (only 1997) were snow measurements, synoptic observations of the board weather station operated by the German Weather Service have been used for R/V Polarstern measurements (period 1995-1997), to decide whether precipitation was of liquid or solid phase (König-Langlo et al., 2006). Measurements on R/V Ron Brown (only 1997) are rain measurements only. Precipitation rates are estimated from disdrometer data as 8 min time series.

A comparison of simultaneous rainfall only measurements on board R/V Alkor, using a disdrometer and a SRG based on 1 minute time series, is given in Fig. 1. The correlation coefficient is 0.9 and the agreement in terms of accumulated rain is excellent, with a deviation of less than 5%.

**2.3 HOAPS**

The satellite derived HOAPS climatology is a compilation precipitation and evaporation data with the goal of estimating the

net freshwater flux from one consistently derived global satellite data set over the global ice-free oceans. To achieve this goal, HOAPS utilizes multi-satellite averages, inter-sensor calibration, and an efficient sea-ice detection procedure. In the utilized version all  HOAPS variables are derived using radiances from the Special Sensor Microwave / Imager (SSM/I) radiometers, except for the sea surface temperature, which is obtained from the Advanced Very High Resolution Radiometer (AVHRR) measurements (Andersson et al., 2010). Three data subsets of HOAPS-3.0 are available over the period 1987 to

2008, comprising scan based pixel-level data (HOAPS-S) and two types of gridded data products (HOAPS-G and HOAPS-C), which make HOAPS useful for a wide range of applications. The HOAPS-S data set, used in the present study, contains all retrieved physical parameters at the native SSM/I pixel-level resolution of approximately 50 km. The HOAPS-3.0 precipitation retrieval is based on a neural network, also described in Andersson et al. (2010).



The detection of very light rain below 0.3 mm h$^{-1}$ is hampered by the sensitivity of the microwave imager. In the HOAPS precipitation algorithm a precipitation signal below the threshold value is set to zero. From experience with the preceding HOAPS precipitation algorithm, a value of 0.3 mm h$^{-1}$ turned out to be an appropriate limit for distinguishing between a real precipitation signal and background noise (Andersson et al., 2010). The algorithm does not discriminate between rain and

snowfall. Due to the strong influence of increasing emissivity near land and sea-ice covered areas, HOAPS is devoid of data within 50 km off any coastline or sea-ice. Therefore ship data within the coastal zones is neglected, too. All individual descending and ascending overpasses of the SSM/I radiometers are used for the ground validation. The position of the HOAPS data represents the centre of an instantaneous field of view, which is about 50 km in diameter.

## 2.4 ECMWF ERA-Interim

Precipitation data is also provided by reanalysis products; here we use data from the ECMWF ERA-Interim reanalysis (Dee et al., 2011). It was initiated in 2006 and represents the latest global atmospheric reanalysis created by the ECMWF. It covers the time period from 1979 onward and is continuously updated on a monthly basis in near-real time. The ERA-Interim reanalysis is produced by means of a 4-D variational data assimilation scheme, which advances forward in time using 12-hourly analysis cycles (Dee et al., 2011). An implemented forecast model estimates the evolving state of the global

atmosphere and surface, and is then constrained by observations of various types and multiple sources, as well as a background estimate of the model. Near-surface parameters are then derived subsequently to upper-air atmospheric fields, both of which serve as a basis for initializing a short-range model forecast that produces a prior state estimate for the successive time step. The short-range forecast, constituting the final part of the ECMWF reanalysis compilation loop, is produced with the Integrated Forecasting System (IFS) (ECMWF, 2006), which comprises a forecast model with three fully

coupled components representing the atmosphere, the land surface, as well as ocean waves. The atmospheric forecast model used for ERA-Interim has a 30 minute time step and a spectral T255 horizontal resolution, which corresponds to roughly uniform 79 km spacing for surface- and other grid point fields (Berrisford et al., 2011). For validation purposes, total precipitation has been extracted from the open-access ECMWF data server. These data consist of surface forecast fields from January 1995 to January 1998 and December 2005 to December 2008, which are initiated at 00 UTC and 12 UTC and

comprise global forecasts with a temporal resolution of three hours. Such short term forecasts were recommended by Kållberg (2011) to be used for validation purposes. The forecast data's temporal resolution is twice as large as that of the surface analysis fields which makes it attractive for validation studies. The ERA-Interim data used within this work are grid point values, meaning that they are not averaged area-wise but are rather valid at the exact location of the grid points (ECMWF, 2015). The grid itself is regular with a 0.75° x 0.75° resolution. Forecast precipitation data is given in the form of

accumulated fields. To obtain the average between two forecast steps, the grid point-wise difference of both single fields was retrieved and multiplied by the inverse of the forecast step.



## 2.5 Simulated precipitation fields

To get an idea of reasonable numbers for the statistical analysis of collocated along-track measurements with areal/temporal averaged estimates, simulations of in-situ observation data sets and corresponding areal averages have been estimated from 8 min time series of precipitation measurements performed on the main building of the GEOMAR in Kiel, Germany. To derive simulated areal averages the scheme given in Fig. 2 has been used based on the assumption that Taylor's principle of frozen turbulence can be applied in a similar manner also to precipitation fields, assuming a speed of motion of 5 m s$^{-1}$ of precipitating clouds. Areal averages $R_{field}$ have been computed according to

$$R_{field} = \sum_{n=1}^{n\,\mathrm{max}} w(n)\, R_{timeseries}(n) \ , \tag{1}$$

where $n$ indicates the n$^{th}$ element of the time series, $w(n)$ is a weighting function according to the number of same measurements $R_{timeseries}(n)$ at a certain time $n$ used for averaging, and normalized by the total number of values used for averaging. In-situ measurements were taken randomly from the same time series. With respect to the different resolutions of HOAPS and ERA-Interim and an assumed displacement of precipitating clouds of 2.4 km within any 8 min interval, we used a 21 x 21 field for HOAPS averaging and a 31 x 31 field for ERA-Interim averaging, which is about 50 km x 50 km for HOAPS and about 75 km x 75 km for ERA-Interim. While for HOAPS a simulated field was estimated at a certain time, for ERA-Interim simulated fields were estimated as an average over 23 consecutive fields in time, each constructed according to Fig. 2, to simulate the forecast increment of 3 hours. This gives nmax=31 for simulated fields of HOAPS and nmax=68 for simulated ERA-Interim fields. For HOAPS-simulations, measurements were taken from the time series within the temporal threshold used for collocation (see chapter 3.1), for ERA-Interim simulated measurements were taken from the members of the time series used for calculating the simulated fields. In case of HOAPS, simulated fields having a precipitation rate below 0.3 mm h$^{-1}$ are set to zero, according to the lower threshold in the HOAPS data.

## 3 Method

The statistical analysis follows the recommendations given by the WMO (World Meteorological Organization) for binary or dichotomous forecasts (WWRP/WGNE, 2014). Therefore it is necessary to collocate the data. In this study precipitation data from 1995-1997, gained over the Atlantic Ocean and the tropical Pacific Ocean, and 2005-2008, gained mainly over the Atlantic Ocean, together yield a point-to-area collocation against the satellite derived climatology HOAPS and ERA-Interim reanalysis data.



### 3.1 Collocation HOAPS

In general there are two different approaches for comparing areal data with in-situ point measurements. The in-situ data can be interpolated to match the position of the satellite or model data, or the satellite/model data is compared to the in-situ measurement that is located closest (nearest neighbour).

As in a similar validation study over the Baltic Sea (Bumke et al., 2012) the second strategy was chosen, because interpolating data has some important disadvantages. To match satellite and in-situ measurements via interpolation, the satellite measurement has to be surrounded by at least 3 in-situ measurements that are spatially and temporally close enough to avoid an impermissible extrapolation. This constraint drastically reduces the number of data suitable for a collocation. Another negative effect of interpolation is smoothing of data, e.g. that minima and maxima are reduced. Thus, the nearest

neighbour approach was chosen. Therefore, it must be ensured that both observations are related to each other, which can be determined by an appropriate decorrelation length. Decorrelation lengths had been derived from in-situ precipitation data over the Baltic Sea: 17 km based on disdrometer 8 min time series (Bumke et al., 2012), 25 km for convective precipitation and 46-68 km for stratiform/frontal precipitation, based on 8 min time series of SRG measurements (Clemens and Bumke, 2002) on board of merchant ships. These numbers give an average decorrelation length of about 30 km, which corresponds

to a temporal correlation length of 45 min assuming a ship's speed of 20 Kn. Therefore, for the HOAPS collocation the allowed time difference was set to 45 min and the allowed distance to 55 km between the ship's position and the satellite's footprint (radius of the SSM/I pixel size plus decorrelation length). These match-up criteria are comparable to allowed differences of 45 min and 50 km chosen in a study by Klepp et al. (2010). According to the satellites' overpass time, collocated data are merged to single events. These last up to 90 min due to the match-up criterions; spatial extension varies

according to the satellites' footprints and ships' positions and speeds (Fig. 3).

### 3.2 Collocation ERA-Interim

Our first requisite is that the time of the measurement is within the forecast interval of 3h, which is the time interval used for accumulation of the predicted precipitation. Furthermore, the four grid points, that enclose the ship's position, were chosen. Again, collocated data are merged to single events, but now according to the time of the observations (1995-1997) or the

time of the forecasts (2005-2008) taking the match-up criterions into account (Fig. 3).

### 4 Statistical Analysis

In the following, the procedure chosen for validation of both HOAPS and ERA-Interim is presented. In total 654 events are available for the period from 1995 to 1997, which are split into 519 rain events and 135 snow events. A total of 2031 collocated rain only events within the second period from 2005 to 2008 for HOAPS and 6011 for ERA-Interim are available.

Each event has been checked as a function of a lower threshold, applied to measured precipitation rates, whether the measurements and reanalysis or satellite data give precipitation. Measured precipitation rates below that threshold were set





to zero. If one or more measured precipitation rates of an event exceed that threshold, the event was flagged as 'observed precipitation yes'; if one of the satellite footprints or the ERA-Interim grid points of an event gives precipitation, independent of its precipitation rate, the event was flagged as 'HOAPS or ERA-Interim precipitation yes'. Therefore, the threshold was set to 0.01 mm h$^{-1}$ for ERA-Interim and all smaller values than this threshold were set to zero.

5 Following the recommendations given by the WMO (World Meteorological Organization) for binary or dichotomous forecasts (WWRP/WGNE, 2014) 2*2 contingency tables are computed. They contain the hits (both HOAPS and measurements or both ERA-Interim and measurements give precipitation), the correct negatives (both HOAPS and measurements or both ERA-Interim and measurements give no precipitation), the misses (observation gives precipitation in contrast to HOAPS/ERA-Interim), and the false alarms (HOAPS/ERA-Interim gives precipitation in contrast to 10 measurements).

This allows us to derive the accuracy, the bias score, the hit rate, and the success ratio. The accuracy is the fraction correct, 1 indicates perfect accuracy. The bias score answers the question: How did the HOAPS/ERA-Interim frequency of "yes" events compare to the observed frequency of "yes" events? A score of 1 means that both data sets include the same number of precipitation events; a larger bias score indicates that there are more precipitation events in the HOAPS or ERA-Interim 15 reanalysis data. The hit rate gives the portion of measured precipitation events, which can also be found in the HOAPS or ERA-Interim reanalysis data, with 1 indicating perfect agreement and 0 no agreement, while the success ratio gives the fraction of precipitation events in HOAPS or ERA-Interim reanalysis data that are also seen in measurements. Again, a score of 1 indicates perfect agreement while 0 indicates maximum disagreement.

## 5 Results

20 The results are given for the earlier period (1995-1997) in terms of accuracy, bias score, hit rate, and success ratio as a function of an assumed lower threshold of measured precipitation rate (Fig. 4), below which the measured precipitation rates were set to zero. The statistical parameters are derived separately for rain events, snow events, and all events; for comparison the results of the simulations (chapter 2.5) are shown. The results for the 2006 to 2008 period are depicted in Fig. 5. The results derived from simulations (chapter 2.5) and the rain only events from the earlier period (1995-1997) are 25 shown for comparison.

Taking into account that ERA-Interim data has no precipitation threshold, only rain rates 1 below 0.01 mm h$^{-1}$ are set to zero for the statistical analysis, best agreement is expected if no threshold is applied to the measurements. For HOAPS, having a lower threshold of 0.3 mm h$^{-1}$, best results can be expected for a lower threshold in measurements similar to that of HOAPS. This is seen in the estimated accuracy, where HOAPS shows the best performance in a range of 0.1 to 0.5 mm h$^{-1}$ threshold 30 in measurements, while ERA-Interim reaches highest values in accuracy when applying no threshold to observational data. Obviously the performance of HOAPS to detect snow is lower compared to ERA-Interim (0.6 to 0.8), but it is still close to the results for simulated data, deviations are less than 0.04 in terms of accuracy for thresholds of 0.1 to 0.5 mm h$^{-1}$ in



measurements. On the other hand, estimated snow accuracies for ERA-Interim decrease rapidly for higher thresholds applied to the measurements. The rain performance of ERA-Interim is weaker than that of HOAPS in terms of accuracy, and for a wide range of thresholds applied to the measurements it is even lower than estimated for simulated fields. Note that the different behaviour of simulations with an increasing lower threshold in the measurements is due to the assumption of a

lower threshold of 0.3 mm h$^{-1}$ in the simulated fields for HOAPS.

The bias score, for which perfect agreement is given at value 1, indicates that ERA-Interim's frequency of precipitation exceeds that of the measurements, strongly increasing with increasing threshold applied to measurements. HOAPS' bias score is close to 1 for a lower threshold in the observation data ranging from 0.1 to 0.2 mm h$^{-1}$, the increase with increasing lower threshold applied to the measurements is less prominent. The results for the simulations indicate that this is mainly a

result of averaging over larger areas due to the resolution of the ERA-Interim analysis, although the model description states that ERA-Interim grid point values do not represent an areal average (ECMWF, 2015).

It is obvious that the hit rate (Fig. 4) should increase with the lower threshold in measurements, because intense measured precipitation events should be easier to detect or to forecast than low intensity precipitation events.  Hit rates for HOAPS are of the order of 0.5 to 0.7; the smallest values are estimated for solid precipitation. Nevertheless, the values of simulations

indicate a good agreement between HOAPS and the measurements. The ERA- Interim precipitation forecasts show hit rates close to 1, not depending on a lower threshold applied to measured data. These high hit rates originate mainly from the high values of the bias score, i.e. when ERA-Interim frequently gives precipitation it also shows precipitation in more or less all cases where precipitation was measured. The hit rate numbers derived from simulations also indicate that this might be partially a result of the coarser spatial resolution.

Finally, the success ratios show reasonable results for both data sets. Success ratios for HOAPS compared to measurements reach values of 0.7 to 0.9 for the lowest threshold applied to the measurements, and decrease with increasing lower threshold, as expected. These numbers are comparable to those of the hit rates and indicate again a good agreement between HOAPS and measurements. Values are a little higher than those estimated from simulations, and differences between different kinds of precipitation are small again. The success ratios of ERA-Interim reanalysis data compared to

measurements are much smaller, which is a consequence of the high frequency of precipitation events in the ERA-Interim reanalysis data as also reflected in the bias score, i.e. when in ERA-Interim forecasts the frequency of precipitation is too high, precipitation is also not measured in all cases.  Nevertheless, the results of the simulations also depict a good performance of the ERA-Interim reanalysis data in correctly predicting the occurrence of precipitation events, taking also into account the coarser resolution of the ERA-Interim reanalysis data. Overall, validation results for solid and liquid

precipitation show a similar level of detection for rain and snow.

Figure 4 gives the results for the second period, 2005 to 2008. For comparison the results for  the period 1995 to 1997, as well as for the simulations are also provided. Since the frequency of events with no precipitation has increased compared to the earlier period, because instruments were running continuously, accuracies have enhanced considerably, but ERA-Interim still gives a less accurate estimate than HOAPS. This is mainly due to an increasing frequency of precipitation events in





ERA-Interim, as indicated also by the bias score, which indicates that the frequency of precipitation in ERA-Interim data is about three times higher than in the measurements and also considerably exceeds the simulated bias score. This is in line with the estimated hit rates, which are again close to 1. The values of the success ratios for HOAPS are close to the estimates of the first period, while ERA-Interim give success ratio numbers that are only about half the values of the earlier periods. In summary, HOAPS performs similarly during both periods while ERA-Interim performs better in the earlier period.

Since SRG and ODM 470 have a very similar performance for measuring and detecting rain (Fig. 1), the main difference between both periods might be caused by the coupling of the SRG and ODM 470 with a rain sensor during the first period. Thus, during the first period, all collocated data are from areas where precipitation is more likely due to the coupling of the instruments with a rain sensor, while during the second period, observational data might cover also areas where precipitation probability is low. To check whether this can explain the differences in the statistical parameters, Figure 6 shows the location of data for both periods separately for HOAPS and ERA-Interim, indicating positions of misses, hits, false alarms, and correct negatives. As can be seen from Fig. 6, for the period 1995 to 1997, there is almost no data from the low-precipitation sub-tropical areas, although e.g. several Atlantic transits of R/V Polarstern were part of the database. This is also in contrast to the data distribution shown for the period 2005 to 2008. Please note in this context that the number of collocated events for ERA-Interim is considerably larger than for HOAPS in 2005 to 2008 as mentioned above. Comparing the number of false alarms (red) and the number of misses (blue) for HOAPS and ERA-Interim, it is obvious that ERA-Interim more frequently gives false alarms compared to HOAPS, while the frequency of misses is very low compared to HOAPS, as supported by estimated bias scores, hit rates, and success ratios. Largest differences are found in the area south of the equator and, less prominently, north of the intertropical convergence zone. Note that Fig. 7 gives the percentage of hits, misses, false alarms, and correct negatives as a function of latitude belts. The most obvious feature is that ERA-Interim gives a much higher number of false alarms and a much lower number of correct negatives. This is supported by a strong increase of the bias score (Fig. 8), which shows a distinct maximum south of the equator, where ERA-Interim gives in terms of frequency 15 times more rain than measured, although the absolute number is relatively small, and a small maximum exists at 10°N-20°N, where ERA-Interim gives about 5-times more frequent rain than observed. The bias score of HOAPS also indicates an overestimation of the frequency of precipitation in the sub-tropics and tropics, but less striking. The latter can also be seen in the context of the number of available events, which are low in the area from 30°S to 10°N (Fig. 8).

Not shown are the results for the Heidke Skill score, which gives the skill of a prediction with respect to a random prediction. The Heidke Skill score shows values close to 0 in the areas between 30°S and the equator for ERA-Interim and between 10°S and the equator and 20°N to 30°N for HOAPS, indicating poor skill of both data sets in these areas compared to observed precipitation, which is reflected also in Fig. 7 and 8. This is also mainly caused by the very small number of observed precipitation events in these areas.

Although the comparison of along-track measurements with areal and temporal averages does not allow to directly compare precipitation rates, the number of events in the first period (657 for HOAPS and ERA-Interim) and in the second period



(2031 for HOAPS and 6011 for ERA-Interim), the latter also including events with no precipitation, allows for a first comparison of average precipitation rates.

Measurements, collocated to HOAPS, of the earlier period give an average precipitation rate of 0.21 mm h$^{-1}$, while HOAPS gives 0.18 mm h$^{-1}$, i.e. an underestimation of 15%. ERA-Interim gives 0.20 mm h$^{-1}$ versus 0.22 mm h$^{-1}$ in collocated measurements of the earlier period, an underestimation of less than 10%.

Using the collocated data pairs and restricting the analysis to non-zero precipitation data, mean values increase to $0.556 \pm 0.122$ mm h$^{-1}$ for measurements versus $0.238 \pm 0.010$ mm h$^{-1}$ for ERA-Interim and $0.578 \pm 0.052$ mm h$^{-1}$ for measurements versus $0.853 \pm 0.041$ mm h$^{-1}$ for HOAPS, where the standard deviation has been estimated using the bootstrap-method (Efron, 1979). The frequency of precipitation is 39.8% for measurements versus 82.6% for ERA-Interim and 36.7% for measurements versus 21.6% for HOAPS.

For the 2005-2008 period, average rain rates from measurements are 0.091 mm h$^{-1}$ versus 0.054 mm h$^{-1}$ from HOAPS data, which means that HOAPS underestimates precipitation considerably (by 40%), and 0.071 mm h$^{-1}$ for measurements versus 0.074 mm h$^{-1}$ for ERA-Interim, which is a slight overestimation of 4% by ERA-Interim. Note that the discrepancies in precipitation rates between both periods are caused by the coupling of the rain gauge and disdrometer with a rain sensor for the earlier period, excluding most events without precipitation.

Using collocated data pairs and restricting the analysis again to non-zero precipitation data, mean values are $0.879 \pm 0.086$ mm h$^{-1}$ for measurements versus $0.169 \pm 0.0013$ mm h$^{-1}$ for ERA-Interim and $0.929 \pm 0.153$ mm h$^{-1}$ for measurements versus $0.821 \pm 0.031$ mm h$^{-1}$ for HOAPS. The frequency of precipitation is 8.1% for measurements versus 43.7% for ERA-Interim and 9.8% for measurements versus 6.5% for HOAPS.

In summary, ERA-Interim overestimates the frequency of precipitation considerably, but combined with lower precipitation rates the mean precipitation is close to measurements. In contrast, HOAPS underestimates the frequency of precipitation, but even the higher average precipitation rate of HOAPS cannot balance the deficit compared to measurements.

For the period 2005-2008, average precipitation rates are also given as a function of latitude for collocated data of measurements and ERA-Interim forecasts (Fig. 9). The main features are a slight underestimation in the mid-latitudes of both, the northern and southern hemispheres, a strong overestimation in the southern sub-tropics and the area of the intertropical convergence zone, and an overestimation in high northern latitudes. Standard deviations were again estimated by applying the bootstrap-method (Efron, 1979) to take into account that precipitation is not Gaussian distributed. They are generally small for ERA-Interim, much smaller than for measurements. Estimates from simulated measurements and fields, both having mean values of 0.19 mm h$^{-1}$, also give smaller standard deviations for the fields (0.028 mm h$^{-1}$) compared to measurements (0.063 mm h$^{-1}$), but differences are not that large. This possibly indicates too little variability in the ERA-Interim precipitation rates. The relatively small number of collocated HOAPS data does not allow for a comparison of precipitation rates with measurements for all latitudinal belts, but north of 40°N and south of 30°S (2005-2008) HOAPS underestimates mean precipitation rates considerably with a total underestimation of about 40%, as mentioned above.





## 6 Discussion

The main problem in interpreting the results is caused by a comparison of along-track point measurements with instantaneous areal estimates (HOAPS) or estimates integrated over time (ERA-Interim). Moreover, the strong spatial and temporal variability and intermittency of the precipitation further complicates the validation efforts. Thus, we cannot expect

a perfect agreement between the different data sets with respect to statistical parameters. To reduce this problem the data have been merged to events (see Fig. 3). In the case of HOAPS each event comprises on average 100 pairs of collocated data, in the case of ERA-Interim, 29 pairs of collocated data for the 1995-1997 period and 69 for the 2005-2008 period. To obtain an idea of reasonable statistical numbers, simulations of point-to-area collocations have been constructed. Comparisons of estimated statistical parameters resulting from these simulated collocations take into account, for example,

that HOAPS data applies a lower precipitation threshold and that ERA-Interim data have a coarser spatial resolution.

Summarized statistical parameters, compared to the results for simulations of point-to-area collocation, show a reasonable ability of HOAPS to detect precipitation. But the results also show that HOAPS underestimates the frequency of precipitation in the mid- and high latitudes, which is partly related to the threshold used in the HOAPS data. This leads to an underestimation in the mean precipitation rate of up to 40%, where the lower threshold in HOAPS may contribute only about

10%, based on statistics from Northern Germany. The ability to detect solid precipitation is a little weaker than for rain, but the derived statistical parameters are still comparable with simulations.

ERA-Interim performs well in areas where precipitation is more likely, especially for snow. Although the bias score is of order 2, it is still of the same order as the simulated data. But the high values of the hit rate, not depending on a lower threshold of measurements, indicate that ERA-Interim overestimates the frequency of precipitation. This is more distinct in

the period 2005 to 2008 and is extreme in the sub-tropical areas. For this period, statistical parameters reveal weaknesses in the ability of ERA-Interim to give correct precipitation forecasts. Nevertheless averaged precipitation rates are close to the measured ones, with biases between -10% and +4% when averaging over all areas. Looking on average precipitation rates for certain areas of the Atlantic Ocean reveals weaknesses. Although the statistics are uncertain in the sub-tropics, where the number of observed precipitation events is very low, results indicate that HOAPS and especially ERA-Interim overestimate

the frequency of precipitation in these areas. For the area of the intertropical convergence zone, both HOAPS and ERA-Interim give too frequent precipitation. While it is not possible to estimate mean precipitation rates in the tropics and sub-tropics due to the low number of observed precipitation events for HOAPS, mean ERA-Interim precipitation rates are higher than observed ones in the inner-tropics. While HOAPS generally  underestimates precipitation rates north of 40°N and south of 30°S (period from 2005-2008), ERA-Interim underestimates precipitation south of 30°S and for latitudes between 40°N

and 50°N, and overestimates it north of 50°N.

The results of this study compare well with results of other studies. Pfeifroth et al. (2012) showed that HOAPS tends to underestimate precipitation by 6 to 9% in the tropical Pacific, where ERA-Interim gives an overestimation of 8 to 10%. Note that these comparisons were done against measurements on atolls, although the representativeness of the atoll gauges of





open-ocean rainfall is still an open question (Wang et al., 2014). Andersson et al. (2011) have shown that HOAPS gives lower precipitation rates than ERA-Interim, except for small areas in the northern sub-tropics, the north-western and south-western Atlantic Ocean. They also figured out that over the mid–high latitudes between 40° and 70° the precipitation (only liquid precipitation) in GPCP (Adler et al., 2003) is also systematically 10%−30% higher relative to HOAPS. Locally the

values exceed 50%. For areas north of 60°N Lindsay et al. (2013) detected an overestimation in monthly precipitation in ERA-Interim ranging from 10 to even 25%. Kidd et al. (2013) found that precipitation in the tropical Atlantic is skewed to the east towards the African coast by ERA-Interim, which might explain the high frequency of precipitation in the data since most measurements in this area are from the eastern tropical Atlantic (Fig. 6). On the other hand, several studies have shown that in the area of the Iberian peninsula (Belo-Pereira et al., 2011) or in the area of four African river basins (Thiemig et al.,

2012) ERA-Interim overestimates the frequency of rain events, as well.

Furthermore, with respect to solid precipitation, Klepp et al. (2010) demonstrated the ability of HOAPS to detect even light amounts of cold season snowfall; with a high accuracy (96%) between point-to-area collocations of ship-based ODM 470s and HOAPS data. Although we don't reach such high accuracies in our analysis, the detectability of solid precipitation in terms of the success ratio shows a good performance in detecting snowfall by both HOAPS and ERA-Interim.

**7 Summary and Outlook**

In this study HOAPS precipitation estimates and ERA-Interim precipitation forecasts have been compared to in-situ SRG and ODM 470 precipitation measurements on board a number of research vessels. The main features are an underestimation in intensity of precipitation by HOAPS, although results have shown that HOAPS performs well in detecting the frequency of precipitation. The frequency of precipitation is strongly overestimated by ERA-Interim, especially in the tropics and sub-

tropics. The Atlantic-wide average precipitation rate is close to measurements, but a distinct feature is an overestimation in the area of the intertropical convergence zone.

These results have to be set into context with evaporation in order to make estimates of the freshwater flux, which influences salinity and is one of the key parameters in ocean modelling. For example, Valdivieso and Haines (2011) figured out that near the surface, much of the Atlantic is generally saltier compared to the climatology, although not uniformly so. They

suspect that precipitation from the ERA-Interim product might have errors. Grist et al. (2014) found that a large part of the spread in the estimates of the mean surface-forced circulation of the sub-tropics is associated with biases in the global ocean heat budgets implied by the atmospheric reanalyses. Since for saltiness an unbiased estimate of freshwater fluxes is needed, good knowledge about evaporation is also essential. Evaporation was the subject of a validation study of Kinzel (2013), who applied a bulk flux parameterization on R/V Polarstern data for the period 1995 to 1997. He showed that HOAPS

underestimates evaporation, except in some areas in the southern Atlantic. Together with an underestimation in precipitation, errors in estimated fresh water fluxes are expected to be small. ERA-interim gives nearly unbiased estimates of evaporation, but regionally there is an overestimation in the tropics and an underestimation at high northern latitudes. The overestimation



of evaporation in the tropics might be balanced by higher precipitation rates according to the present study, but in higher latitudes it is not the case.

So in the course of 2016, when HOAPS 4 data are also available for 2008 onwards, an extended time period will be used to enhance the validation study for precipitation and to also include evaporation in order to make direct estimates of the freshwater flux. In this context, it also has to be pointed out that the other main problem in validating precipitation over the sea (beside the problem comparing point measurements with areal estimates) is the paucity of accurate precipitation measurements over sea. Extending the investigation for the time after 2008 will make the results more robust against single events. It is also planned, in cooperation with OceanRAIN (Klepp, 2015), to use another set of precipitation data gained by ODM 470s mounted on 9 research vessels worldwide. These data cover also the period after 2008 and will put the validation on a broader basis.

**Data availability**

Most of the data are accessible from data bank systems. Information is given in the data description in chapter 2. Precipitation measurements of the 1995-1997 period, measurements on R/V Alkor and on the GEOMAR building in Kiel will be archived in the Pangaea data library.

**Acknowledgements**

We would like to thank especially the masters and their crews, for supporting the precipitation measurements on board all research vessels. Meteorological data of R/V Maria S. Merian are provided from the DOD (Deutsches Ozeanographisches Datenzentrum) maintained by the Bundesamt für Seeschifffahrt und Hydrographie. Special thanks go to Reimer Wolf from Briese for his engagement in performing precipitation measurements on R/V Maria S. Merian and to Lutz Hasse, who initiated the precipitation measurements over the sea, and his former working group in Kiel. ERA-Interim data used in this study have been obtained from the ECMWF data server. Our thanks go also to Christian Klepp and to Lisa Neef for proof reading.



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

Figure 1. Comparison of simultaneous measurements of precipitation (rain only) on board R/V Alkor in the Baltic Sea (data from 1999 – 2005, May to October) using an ODM 470 and a SRG. Interval of measurements is 1 minute.



Figure 2. Scheme for estimates of simulated precipitation fields from precipitation time series gained on the main building of the GEOMAR in Kiel (°N, °E). The numbers indicate the $n^{th}$ 8 min interval of the time series. Left: Scheme for a spatial field of 75 km x 75 km representing ERA-Interim resolution at the beginning of a 184 min period, right at the end of this period. An ERA-Interim grid point is simulated from 23 consecutive fields. The X indicates an observation, which is taken randomly from elements n = 1 to n = 68 of the time series for ERA-Interim simulations (elements n = -5 to n = 37 for HOAPS simulations).

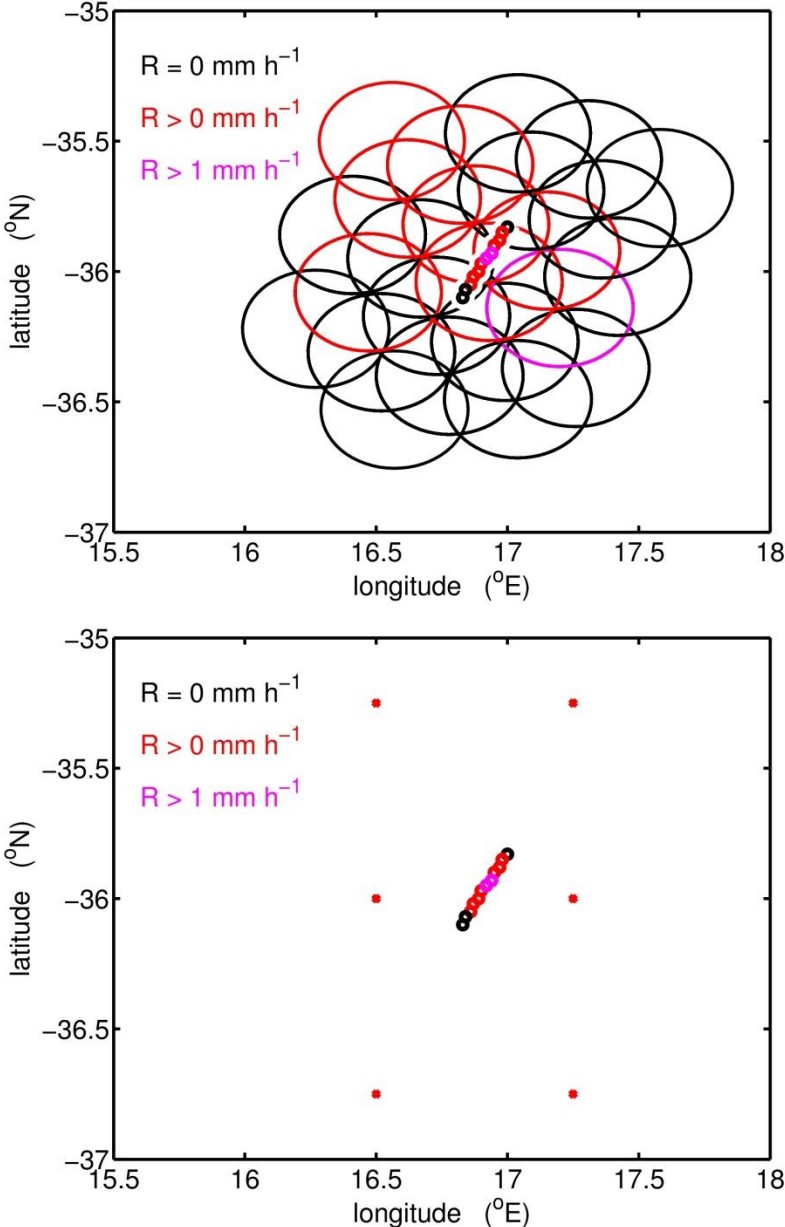

Figure 3. Examples of collocated data, which have been merged to events for HOAPS (top) and ERA-Interim (bottom). HOAPS' footprints are indicated by the large circles, observations by o, and ERA-Interim gridpoints by ●. Colors indicate the rain rates R. HOAPS data are from day 338 in 1995, minute 1272, collocated observations are from minutes 1235 to 1315. In case of ERA-Interim, forecasts are from day 338, minutes 1260 and 1440, collocated observations are from day 338, minutes 1227 to 1315.





Figure 4. Accuracy, bias score, hit rate, and success ratio (from top to bottom) against precipitation measurements for the period 1995-1997 (left HOAPS, right ERA-Interim) as a function of a lower threshold applied to the measurements.







Figure 5. Accuracy, bias score, hit rate, and success ratio (from top to bottom) against precipitation measurements for the period 2005-2008 (left HOAPS, right ERA-Interim) as a function of a lower threshold applied to the measurements.

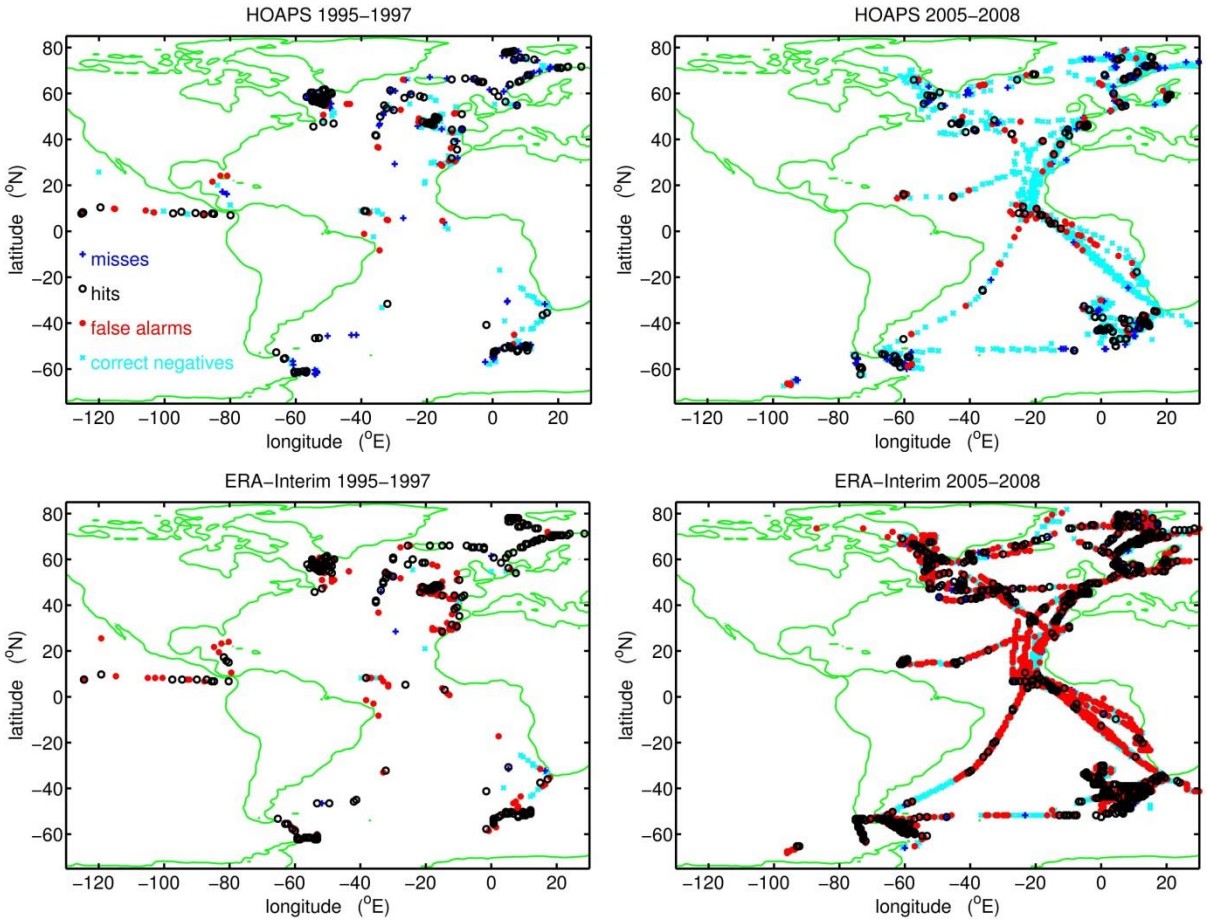

Figure 6. Location of collocated events (left: HOAPS and measurements, right: ERA-Interim and measurements) for the period 1995-1997 (top) and 2005-2008 (bottom). The different coloured symbols indicate misses, hits, false alarms, and correct negatives. A lower threshold is not set.



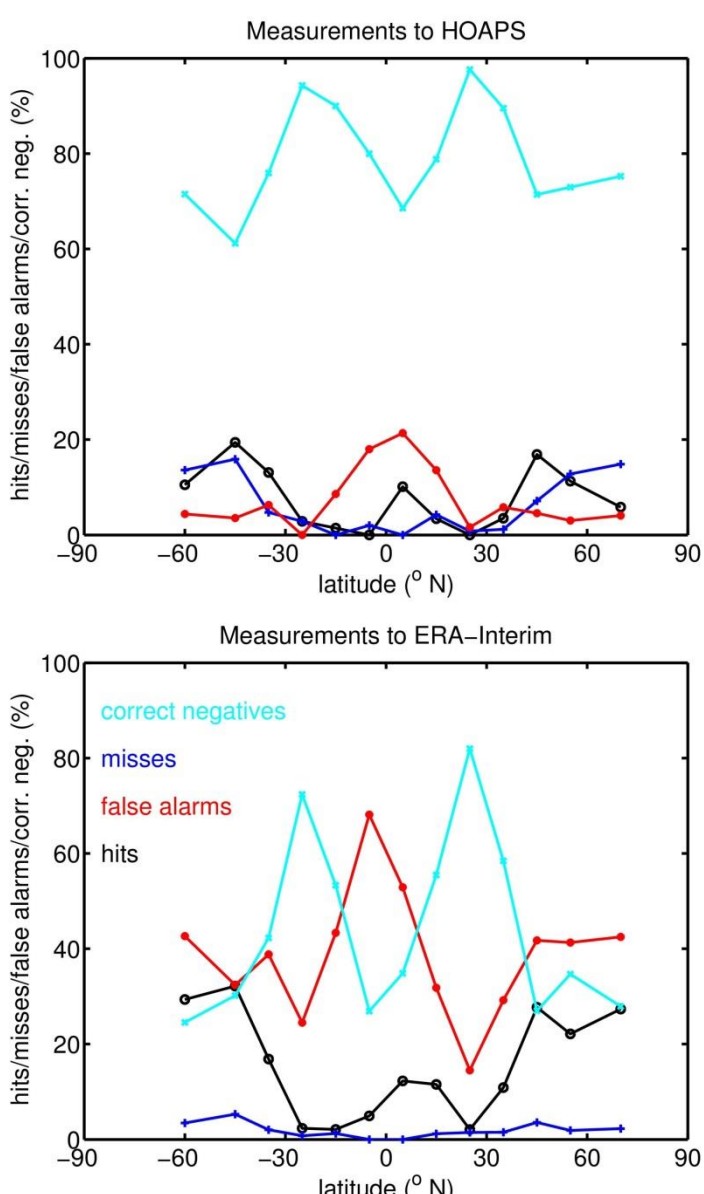

Figure 7. Hits, misses, false alarms, and correct negatives of events for the period from 2005-2008 as a function of latitude (latitude belts are south of 50°S, 50°S − 40°S, 40°S-30°S, 30°S-20°S, 20°S-10°S, 10°S-0°, 0°-10°N, 10°N-20°N, 20°N-30°N, 30°N-40°N, 40°N-50°N, 50°N-60°N, and north of 60°N) for HOAPS (top) and ERA-Interim (bottom). A lower threshold is not set for ERA-Interim data.





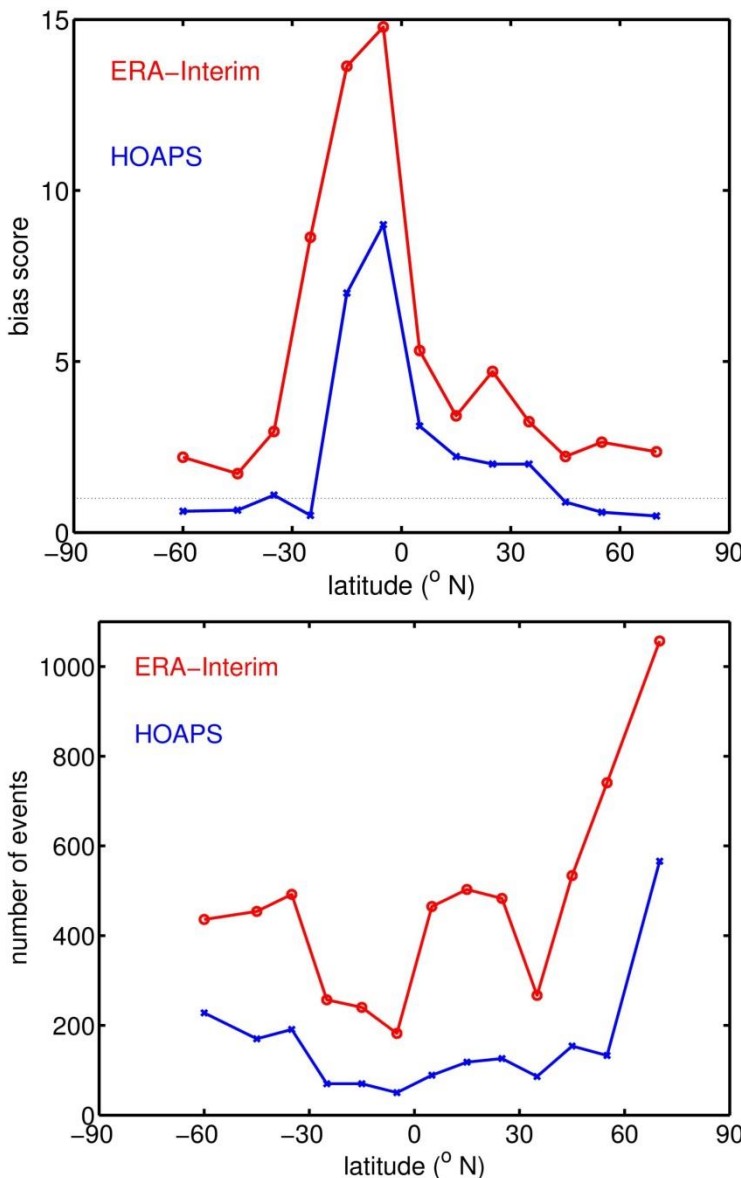

Figure 8. Bias score (top) and number of events (bottom) for the period 2005-2008 as a function of latitude (latitude belts are south of 50°S, 50°S – 40°S, 40°S-30°S, 30°S-20°S, 20°S-10°S, 10°S-0°, 0°-10°N, 10°N-20°N, 20°N-30°N, 30°N-40°N, 40°N-50°N, 50°N-60°N, and north of 60°N) for HOAPS and ERA-Interim. A lower threshold is not set for ERA-Interim data.

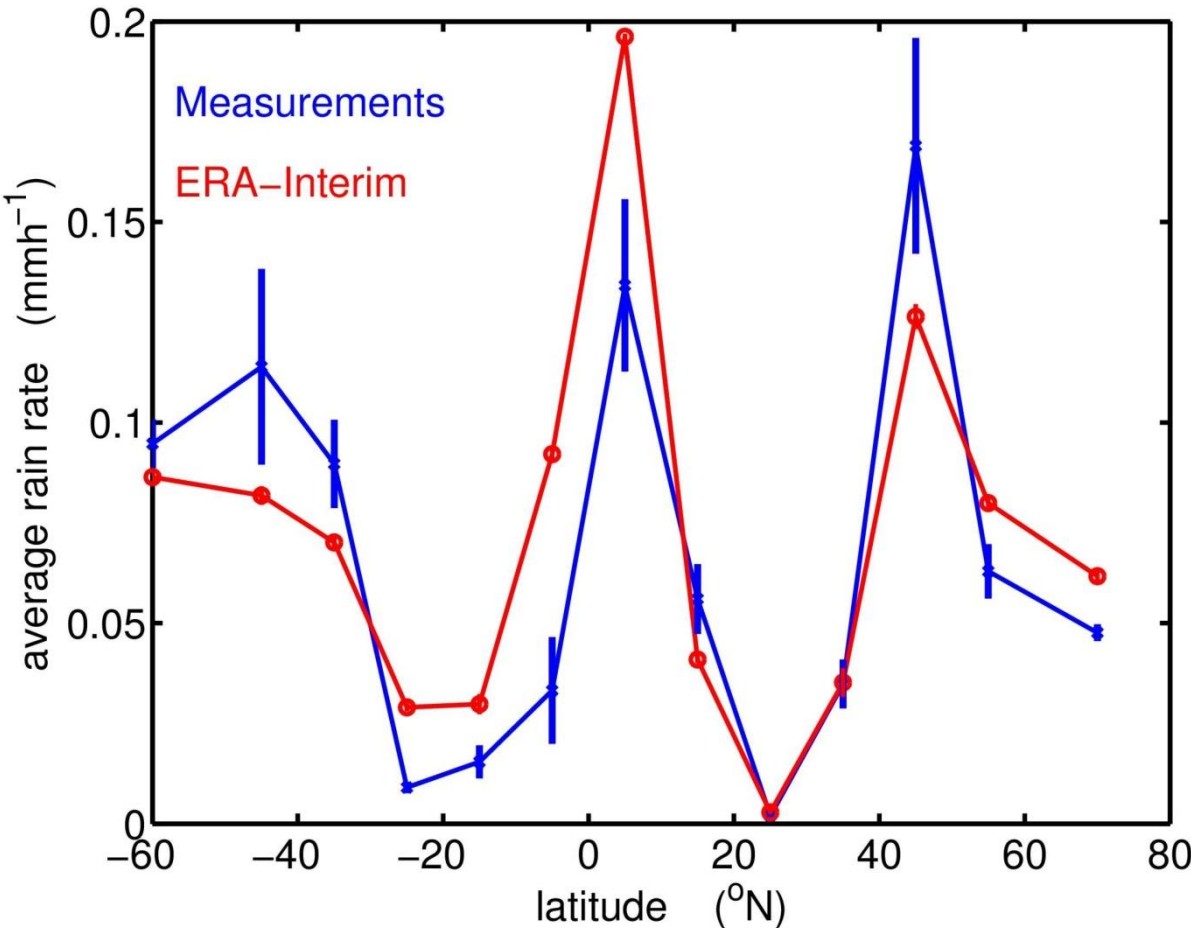

Figure 9. Mean precipitation rates of collocated data pairs for the period from 2005-2008 as a function of latitude (latitude belts are south of 50°S, 50°S – 40°S, 40°S-30°S, 30°S-20°S, 20°S-10°S, 10°S-0°, 0°-10°N, 10°N-20°N, 20°N-30°N, 30°N-40°N, 40°N-50°N, 50°N-60°N, and north of 60°N) for ERA-Interim compared to collocated measurements. Vertical bars indicate standard-deviation estimated by applying the bootstrap method (Efron, 1979). A lower threshold is not set for ERA-Interim data.