# Peer review of "HOAPS and ERA-Interim precipitation over sea: Validation against shipboard in-situ measurements"

_Atmospheric Measurement Techniques, 2016_

## Referee Comment (RC1) · Anonymous Referee #1 · 24 Feb 2016

The authors have compared two precipitation data sets, HOAPS and ERA-Interim with shipboard gauge and disdrometer measurements for the spans 1995-1997 and 2005-2008. I find shipboard precipitation measurements potentially valuable as there is little in-situ information about precipitation over the open ocean. As the authors' state, low lying atolls are often used as a proxy for open ocean. While I do find merit to this work and the manuscript is comprehensive, I found the sentence structure choppy and very difficult to follow. Hence, my comments are limited to overall results. My hope is that if/when the manuscript is edited it will flow better and not be quite as difficult to read. Detailed comments are provided below.

(1) I highly suggest the authors' summarize the results in tabular form or at least focus

[Figure]

of the most important results. Stating every result sentence by sentence is tedious to read and detracts from the readability. For example, the results on Page 11, lines 1-19, could be listed in tabular form. It's quite taxing to read and comprehend in an efficient manner.

(2) Being a satellite data person, I find the HOAPS comparisons quite interesting and illuminating. Satellite data sets are generally more challenging to characterize than the more predictable behavior of model and reanalysis data. I feel the manuscript would be much more interesting and useful if additional satellite-based precipitation data sets were compared with the shipboard gauges. I realize this is significant additional work, but comparisons with the shipboard gauges could be quite useful for ocean validation purposes.

(3) I believe it's well-known that ERA-Interim overestimates precipitation frequency so the authors' could have forecast this result. I'm not quite sure how useful it is dwelling on this, but I understand the authors' need to be thorough.

(4) Page 2, Line 21 - "male-functions" should be "malfunctions".

(5) Page 9, Line 31 - "Figure 4" should be "Figure 5".
* * *

---

## Referee Comment (RC2) · Anonymous Referee #2 · 23 Mar 2016

The paper "HOAPS and ERA-Interim precipitation over sea: Validation against shipboard in-situ measurements" by K. Bumke and co-workers presents a validation study to assess the performances of two precipitation datasets (HOAPS and ERA-I) over ocean, after comparisons with ship measurements at global scale. The objective is challenging, since instantaneous areal precipitation (HOAPS) and grid-point cumulated precipitation (ERA-I) have to be matched with time integrated point-like measurements from moving platforms. The paper is interesting and presents significant results, deserving the publication on AMT. However, I recommend a moderate revision, focused on improving presentation and increasing the information content of the results.

Abstract. Line 11 (and throughout the paper), none of the two products validated in this

work are strictly "forecasts": I suggest to refer to the validated fields as "estimates" or "products" instead.

Introduction. This section has to be improved. First, no mention of the most popular satellite derived precipitation products is given. Global products such as TMPA (Huffman et al., 2007. J. Hydrometeor.,8, 38–55), C-MORPH (Joyce et al. 2004, J. Hydrometeor., 5, 487–503) PERSIANN (Behrangi et al. 2009, J. Hydrometeor., 10, 1414–1429), H-SAF (Mugnai et al., 2013, NHESS, 13, 1959–1981) and the newest IMERG (Huffmann et al. 2015, http://pmm.nasa.gov/sites/default/files/document_files/IMERG_doc.pdf, should be at least mentioned. Moreover, the introductory discussion of literature about evaporation (now on lines 23-30 on page 13) and HOAPS/ERA-I validation (lines 31-page 12 to line 14-page 13) should be moved here.

Section 2. Page 2. Here there is no real reason to mention the exact position of the data and I suggest not mentioning Figure 6 (it is also repeated in similar sentences at lines 24 and 28). Page 3, lines 9-10. Later in the paper, it is stated that that the ships' speed is 20 Kn. This information should be anticipated here, indicating if this value is a kind of average of the cruise speed or it is just an order of magnitude. Figure 2 is confusing, and does not help in understanding section 2.5. I suggest to better explaining all the features in the figure (e.g. what is the meaning of the colors?), to improve the caption and to improve the clarity of section 2.5.

Section 3. Page 7, lines 6-9. The discussion on the use of an interpolation system for rainrate data is probably out of the scope of the paper (many interpolators do not affect rainrate maxima and minima). I suggest to simply say that nearest neighbor is used following Bumke et al., 2012. Page 7, line 14. It is 30 km a weighted mean of the decorrelation distance? How it is computed? Since the data domain ranges over all latitudes and seasons, it would be possible (and useful) to apply different correlation lengths, according to the precipitation type (convective/stratiform)?

Section 4. Page 8, line 11. About the considered indices, I suggest the following. 1) avoid to use the "accuracy": it depends on the number of correct negatives, and this number varies with the data acquisition strategy, affecting the results; 2) replace accuracy with some equitable skill score, such as the ETS or HSS (mentioned below in the manuscript) to assess the skill of the product with respect to random rain assignments; 3) rename hit rate with Probability of Detection (POD), and write explicitly that the success ratio can be considered as 1-FAR (False Alarm Ratio): POD and FAR are more self-explaining (and popular) names for these indicators (see, among others, Puca etal., 2014, NHESS, 14, 871-889).

Section 5. The discussion on the thresholds (Figs 4, 5) should include the analysis of the number of "wet" events in the database, that are expected to decrease rapidly with increasing thresholds, since rainrate should be distributed as a power-low. This is the reason I suggest using equitable skill scores instead of accuracy.

Section 6. The second part of this section (below line 31 on page 12) is a review of literature results and should go in the Introduction.

Section "Data availability". I suggest to cancel this unnumbered section and to include data providers in the Acknowledgement section.

---

## Author Comment (AC1) · 4 Apr 2016

(1) I highly suggest the authors' summarize the results in tabular form or at least focus of the most important results. Stating every result sentence by sentence is tedious to read and detracts from the readability. For example, the results on Page 11, lines 1-19, could be listed in tabular form. It's quite taxing to read and comprehend in an efficient manner.

1) This part (p 10, starting at line 32) has been rewritten and Tab.2 has been added.

Although the comparison of along-track measurements with areal and temporal averages does not allow for directly comparing precipitation rates, the number of events in the first period (657 for HOAPS and ERA-Interim) and in the second period (2031 for HOAPS and 6011 for ERA-Interim), the latter also including more events with no precipitation, allows for a first comparison of average precipitation rates (Table 2a and b).

In the earlier period, where precipitation is more likely, HOAPS underestimates observed precipitation by 15% and ERA-Interim by less than 10%. For the 2005-2008 period, HOAPS underestimates precipitation considerably by 40%, while ERA-Interim performs much better with a slight overestimation of 4%, only (Table 2a).

Using collocated data pairs instead of precipitation events and restricting the analysis to non-zero precipitation data, the number of data is sufficient to estimate the standard deviation by applying the bootstrap-method (Efron, 1979). For the 1995-1997 period HOAPS shows in average significantly higher and ERA-Interim significantly lower precipitation rates than observed (Table 2b). These differences in precipitation rates were in parts balanced by precipitation frequency, which is lower for HOAPS and considerably higher for ERA-Interim than observed. For the 2005-2008 period ERA-Interim shows again significantly lower precipitation rates than observed, while HOAPS compares well with observations, the deviation is within a standard deviation. However, the deficit in the average precipitation rate of ERA-Interim goes along with an extreme overestimation of precipitation frequency of more than 400%, while HOAPS shows an underestimation of precipitation frequency by about one third (Table 2b and c).

In summary, ERA-Interim overestimates the frequency of precipitation considerably, but combined with low precipitation rates the mean precipitation is close to measurements. In contrast, HOAPS underestimates the frequency of precipitation, but even the higher average precipitation rate of HOAPS cannot balance the deficit compared to measurements.

a)

| Period | Average precipitation rate (mm/h) | | | |
| --- | --- | --- | --- | --- |
| | Measurements coll. to HOAPS | HOAPS | Measurements coll. to ERA-I. | ERA-Interim |
| 1995-1997 | 0.21 | 0.18 | 0.22 | 0.20 |
| 2005-2008 | 0.091 | 0.054 | 0.071 | 0.074 |

b)

| Period | Average precipitation rate and standard deviation (mm/h) | | | |
|---|---|---|---|---|
| | Measurements coll. to HOAPS | HOAPS | Measurements coll. to ERA-I. | ERA-Interim |
| 1995-1997 | 0.578±0.052 | 0.853±0.041 | 0.556±0.122 | 0.238±0.010 |
| 2005-2008 | 0.929±0.153 | 0.821±0.031 | 0.879±0.086 | 0.169±0.0013 |

c)

| Period | Precipitation frequency (%) | | | |
|---|---|---|---|---|
| | Measurements coll. to HOAPS | HOAPS | Measurements coll. to ERA-I. | ERA-Interim |
| 1995-1997 | 36.7 | 21.6 | 39.8 | 82.6 |
| 2005-2008 | 9.8 | 6.5 | 8.1 | 43.7 |

Table 2: Average precipitation rates of precipitation events (a) and average precipitation rates and their standard deviation for all collocated data pairs restricted to non-zero precipitation data (b) for collocated measurements and collocated HOAPS respective ERA-Interim data. (c) gives the precipitation frequencies derived from all collocated data pairs of both data sets.

(2) Being a satellite data person, I find the HOAPS comparisons quite interesting and

illuminating. Satellite data sets are generally more challenging to characterize than the

more predictable behavior of model and reanalysis data. I feel the manuscript would

be much more interesting and useful if additional satellite-based precipitation data sets

were compared with the shipboard gauges. I realize this is significant additional work,

but comparisons with the shipboard gauges could be quite useful for ocean validation

purposes.

We agree that it would be desirable to validate also other precipitation products. In this context the present study can be seen as a kind of a pilot study to investigate the possibility to validate areal or spatial averaged precipitation data sets with along track onboard measurements of precipitation by using ship rain gauges and optical disdrometers. Therefore we have chosen HOAPS as an example for a high resolution , necessary for collocation purposes, remote sensing data set and ERA-Interim as a representative data set for reanalysis data. Furthermore HOAPS is derived only from one sensor and given in its native resolution without any interpolation. In future it is planned to extend as well the data base, within the project Ocean Rain (Klepp et al., 2015) a number of research vessels have been equipped with optical disdrometers in recent years, as the number of data sets used to be validated.

2) I believe it's well-known that ERA-Interim overestimates precipitation frequency so the
authors' could have forecast this result. I'm not quite sure how useful it is dwelling on this,
but I understand the authors' need to be thorough.

You are right, indeed we forecasted in some sense this result. What we did not forecast was that the high frequencies of rain balance more or less the low rain rates and, moreover, the latitudinal differences, which, in our opinion, justify "dwelling on this".

3) Page 2, Line 21 - "male-functions" should be "malfunctions".

Is corrected

4) Page 9, Line 31 - "Figure 4" should be "Figure 5".

Is corrected

Furthermore we tried to improve the English.

---

## Author Comment (AC2) · 4 Apr 2016

Abstract. Line 11 (and throughout the paper), none of the two products validated in this work are strictly "forecasts": I suggest to refer to the validated fields as "estimates" or "products" instead.

We changed the text accordingly.

Introduction. This section has to be improved. First, no mention of the most popular satellite derived precipitation products is given. Global products such as TMPA (Huffman et al., 2007. J. Hydrometeor.,8, 38–55), C-MORPH (Joyce et al. 2004, J. Hydrometeor., 5, 487–503) PERSIANN (Behrangi et al. 2009, J. Hydrometeor., 10, 1414–1429), H-SAF (Mugnai et al., 2013, NHESS, 13, 1959–1981) and the newest IMERG (Huffmann et al. 2015, http://pmm.nasa.gov/sites/default/files/document_files/IMERG_doc.pdf, should be at least mentioned. Moreover, the introductory discussion of literature about evaporation (now on lines 23-30 on page 13) and HOAPS/ERA-I validation (lines 31-page 12 to line 14-page 13) should be moved here.

We followed the suggestions of the reviewer and revised the introduction as well as page 12 and page 13. The revised introduction is given below:

[revised manuscript text omitted]

In the present study we use in-situ ship rain gauge (Hasse et al., 1998) and optical disdrometer (ODM 470) data (Großklaus et al., 1998), gained on board of research vessels, to validate two data sets, the HOAPS and the ECMWF (European Centre for Medium-Range Weather Forecasts) ERA-Interim reanalysis data set. For the present study HOAPS has been chosen as an example of a satellite derived precipitation estimate, because it is only derived from SSM/I (Special Sensor Microwave Imager) radiometers in their native sensor resolution without any further interpolation.

Section 2 gives an overview over the data and instruments used. The collocation of the data is described in section 3, followed by an overview of used validation methods in section 4. In section 5 we present our results, followed by a discussion of the results and the summary and outlook in sections 6 and 7.

Section 2.
Page 2. Here there is no real reason to mention the exact position of the
data and I suggest not mentioning Figure 6 (it is also repeated in similar sentences
at lines 24 and 28).

We deleted mentioning Fig. 6

Page 3, lines 9-10. Later in the paper, it is stated that that the
ships' speed is 20 Kn. This information should be anticipated here, indicating if this
value is a kind of average of the cruise speed or it is just an order of magnitude.

Cruise speed is that of the merchant ships over the Baltic Sea to relate temporal and spatial scales, we changed the sentence to:
A decorrelation length of 30 km, which corresponds to a temporal correlation length of 45 min assuming a merchant ship's speed to be of the order of 20 Kn.

Figure 2 is confusing, and does not help in understanding section 2.5. I suggest to better
explaining all the features in the figure (e.g. what is the meaning of the colors?), to
improve the caption and to improve the clarity of section 2.5.

In fact the colors indicate only same elements of the time series used to derive simulated fields, which is now mentioned in the capture to Fig.2:

Figure 2. Scheme for estimates of simulated precipitation fields from precipitation time series gained on the main building of the GEOMAR in Kiel (54.3° N, 10.2° E) at a time interval of 8 min. The numbers, additionally highlighted by colours, indicate the nth 8 min interval of the time series. Assuming a speed of motion of precipitating clouds of 5 m s-1, each 8 min interval is equivalent to a displacement of 2.4km. Thus, a spatial field of 75 km x 75 km is simulated by 31 elements x 31 elements of the 8 minutes time series. Since ERA-Interim precipitation is accumulated over 3 h, simulated data have to be averaged over 23 consecutive fields (23·8 min=184 min). Left: Scheme for a spatial field representing ERA-Interim resolution at the beginning of a 184 min period, right at the end of this period. Start elements are indicated by the red colour. The X indicates an observation, which is taken randomly from elements n = 1 to n = 68 of the time series for ERA-Interim simulations and from elements n = -5 to n = 37 for HOAPS simulations due to collocation criteria (chapter 3.1 and 3.2).

Section 2.5 has been changed to improve the clarity:

To get an idea of reasonable numbers for the statistical analysis of collocated along-track measurements with areal/temporal averaged estimates, simulations of in-situ observation data sets and corresponding areal averages have been estimated from 8 min time series of precipitation measurements performed on the main building of the GEOMAR in Kiel, Germany. To derive simulated areal averages from a time series the scheme given in Fig. 2 has been used based on the assumption that Taylor's principle of frozen turbulence can be applied to

precipitation fields in a similar manner, assuming a speed of motion of 5 m s$^{-1}$ of precipitating clouds. Thus, an 8 min interval is equivalent to a displacement of precipitating clouds by 2400 m. Areal averages $R_{field}$ have been computed according to

$$R_{field} = \sum_{n=1}^{n\,max} w(n)\, R_{timeseries}(n) \; , \tag{1}$$

where $n$ indicates the $n^{th}$ element of the time series, $w(n)$ is a weighting function according to the number of same elements of the time series $R_{timeseries}(n)$ at a certain time $n$ used for averaging, normalized by the total number of values used for averaging. Simulated in-situ measurements were taken randomly from the same time series. With respect to the different resolutions of HOAPS and ERA-Interim and the assumed displacement of 2.4 km within any 8 min interval, we used a 21 x 21 field (21·8·60 s·5 m s$^{-1}$=50.4 km) for HOAPS averaging and a 31 x 31 field for ERA-Interim averaging. (31·8·60 s·5ms$^{-1}$=74.4km). While for HOAPS a simulated field was estimated at a certain starting time, simulated fields for ERA-Interim were estimated as an average over 23 consecutive fields in time (23·8 min=184 min), each constructed according to Fig. 2, to simulate the time increment of 3 hours. Starting with element n=1 this gives nmax=31 for simulated fields of HOAPS and nmax=68 for simulated ERA-Interim fields. For HOAPS-simulations, simulated point measurements were taken from those elements of the time series, which are within the temporal threshold used for collocation (see chapter 3.1), for ERA-Interim simulated measurements were taken from all members of the time series used for calculating the simulated fields. In case of HOAPS, simulated fields having a precipitation rate below 0.3 mm h$^{-1}$ are set to zero, according to the lower threshold in the HOAPS data.

Section 3. Page 7, lines 6-9. The discussion on the use of an interpolation system
for rainrate data is probably out of the scope of the paper (many interpolators do not
affect rainrate maxima and minima). I suggest to simply say that nearest neighbor is
used following Bumke et al., 2012.

We deleted the first paragraph and started with:
As in a similar validation study over the Baltic Sea (Bumke et al., 2012), the nearest neighbour approach was chosen for collocation. Therefore, it must be …

Page 7, line 14. It is 30 km a weighted mean of the
decorrelation distance? How it is computed?

It is simply an average value from different estimates of the decorrelation lengths based on 8 min timeseries, where we assumed the lower decorrelation length for stratiform/frontal precipitation;
(17km + 25km + 46km) : 3 = 88km : 3 ≈ 30km
We changed the text and added Puca et al. an additional reference:

Using 46 km as decorrelation length for stratiform/frontal precipitation these three numbers give an average decorrelation length of about 30 km which agrees well with a study of Puca et al., 2014. A decorrelation length of 30 km corresponds to a temporal correlation length of 45 min assuming a merchant ship's speed to be of the order of 20 Kn.

Since the data domain ranges over all
latitudes and seasons, it would be possible (and useful) to apply different correlation
lengths, according to the precipitation type (convective/stratiform)?

Unfortunately the data do not include any information about the precipitation type. So we used an average decorrelation length for all collocations, derived for 8 min time series. On the other hand we merged the data to events, thus, each event represents, also with respect to measurements, a temporal/spatial average. In fact measurements of collocated HOAPS events represent an average over about 1 hour. Indeed this means that decorrelation lengths also should increase, see for example (http://postel.obs-mip.fr/IMG/pdf/CSP-0350-ATBD_Precipitation-I2.00.pdf). That the chosen decorrelation lengths are uncritical is supported by test runs, where we changed decorrelation lengths for collocation purposes by +-10km. This had no significant influence on the statistics.

Section 4. Page 8, line 11. About the considered indices, I suggest the following. 1)

avoid to use the "accuracy": it depends on the number of correct negatives, and this
number varies with the data acquisition strategy, affecting the results; 2) replace accuracy
with some equitable skill score, such as the ETS or HSS (mentioned below in
the manuscript) to assess the skill of the product with respect to random rain assignments;
3) rename hit rate with Probability of Detection (POD), and write explicitly that
the success ratio can be considered as 1-FAR (False Alarm Ratio): POD and FAR
are more self-explaining (and popular) names for these indicators (see, among others,
Puca etal., 2014, NHESS, 14, 871-889).

In the submitted version, although not written explicitly, we stated that changes in the accuracy between both periods, was caused by differences in the data acquisition scheme, i.e. the coupling with a rain sensor. Instead of deleting the results of the accuracy we mention now clearly that it depends also on correct negatives and that it can be considered as 1-FAR as suggested. We added now figures for the HSS and changed everything accordingly:

Starting with the last paragraph of chapter 4:
This allows us to derive the accuracy, the bias score, the probability of detection (POD), the success ratio and the Heidke skill score. The accuracy is the fraction correct, 1 indicates perfect accuracy. It also depends on the number of correct negatives. The bias score answers the question: How does the HOAPS/ERA-Interim frequency of "yes" events compare to the observed frequency of "yes" events? A score of 1 means that both data sets include the same number of precipitation events; a larger bias score indicates that there are more precipitation events in the HOAPS or ERA-Interim reanalysis data. The POD gives the portion of measured precipitation events, which can also be found in the HOAPS or ERA-Interim reanalysis data, with 1 indicating perfect agreement and 0 no agreement, while the success ratio gives the fraction of precipitation events in HOAPS or ERA-Interim reanalysis data that are also seen in measurements. Again, a score of 1 indicates perfect agreement while 0 indicates maximum disagreement. It is equal to 1 minus the false alarm ratio, where the false alarm ratio gives the fraction of the observed no-events which were incorrectly forecasted as yes. The Heidke skill score measures the fraction of correct estimates after eliminating those which would solely be correct by random chance. 0 indicates no skill, perfect score equals to 1.

**5 Results**

[revised manuscript text omitted]

Section 5. The discussion on the thresholds (Figs 4, 5) should include the analysis of the number of "wet" events in the database, that are expected to decrease rapidly with increasing thresholds, since rainrate should be distributed as a power-low. This is the reason I suggest using equitable skill scores instead of accuracy.

Table 1 has been added giving the numbers of wet events, see above.

Section 6. The second part of this section (below line 31 on page 12) is a review of literature results and should go in the Introduction.

Went into the introduction, please see above.

Section "Data availability". I suggest to cancel this unnumbered section and to include data providers in the Acknowledgement section.

Is now included in the acknowledgements